# First Report of Genetic Resistance to Azithromycin in *Treponema pallidum* from Blood Samples Among Men Who Have Sex with Men and People Living with HIV from Mexico

**DOI:** 10.3390/microorganisms13051069

**Published:** 2025-05-03

**Authors:** Dayana Nicté Vergara-Ortega, Perla J. Santibañez-Amador, Santa García-Cisneros, María Olamendi-Portugal, Everardo Gutiérrez-Millán, Antonia Herrera-Ortíz, Verónica Ruíz-González, Miguel Ángel Sánchez-Alemán

**Affiliations:** 1Centro de Investigación Sobre Enfermedades Infecciosas, Instituto Nacional de Salud Pública, Cuernavaca CP 62100, Mexico; nicte.vergara@gmail.com (D.N.V.-O.); perla.santibanez@uaem.edu.mx (P.J.S.-A.); sgarcia@insp.mx (S.G.-C.); molamendi@insp.mx (M.O.-P.); ever.gmillan@gmail.com (E.G.-M.); aherrera@insp.mx (A.H.-O.); 2Laboratorio Especializado, Clínica Especializada Condesa, Cuauhtémoc CP 06170, Mexico; vruiz@gmail.com

**Keywords:** syphilis, macrolides, azithromycin, drug resistance, men who have sex with men, people living with HIV

## Abstract

Syphilis is a re-emerging sexually transmitted disease caused by *Treponema pallidum* subsp. *pallidum* (TPA). It especially affects vulnerable populations such as men who have sex with men (MSM) and people living with HIV. Despite being treatable with benzathine penicillin G, a substantial increase in TPA resistance to azithromycin has been reported in many countries. The objective of this study was to detect the resistance of *T. pallidum* (TPA) to macrolides in blood samples from men who have sex with men and people living with HIV using molecular methods in a cross-sectional study. The detection of both TPA and the resistance to azithromycin was achieved through molecular methodologies (nested PCR), which were applied to blood samples of people with asymptomatic syphilis. We report the first data on the molecular prevalence of TPA and the first identification of genetic resistance to azithromycin (punctual mutation A2058G) in Mexico. Resistance testing for syphilis is not routinely performed in Mexico, but azithromycin continues to be prescribed despite syphilis being treatable with benzathine penicillin G. Therefore, the surveillance of cases of syphilis treatment failure, especially in vulnerable populations, which are the population group that maintains the active transmission of TPA, is recommended.

## 1. Introduction

Syphilis is an ancient sexually transmitted disease caused by the bacterium *Treponema pallidum* subsp. *pallidum* (TPA), and it is currently in re-emergence. Since 2000, the number of reports of both congenital and acquired syphilis cases has increased significantly in many countries [1,2]. This public health problem affects vulnerable populations, especially men who have sex with men (MSM) and people living with HIV (PLWH); MSM constitutes the core group responsible for syphilis transmission [3].

With respect to the molecular diagnosis of syphilis, diverse studies have revealed an improvement in the sensitivity of conventional PCR using ulcer samples because of their higher TPA bacterial load in comparison to blood, plasma, serum, and urine. The disadvantage is that the window period for active ulcers is short. Similarly, this window of active ulcers may overlap with another period ranging from 3 to 12 weeks, during which *T. pallidum* undergoes hematogenous and lymphatic dissemination. These phases have been described as optimal for sample collection and the molecular detection of the bacterium [4,5]. Several genes have been proposed as targets for detecting TPA in clinical samples, but the most commonly used genes are *Tpn47* and *polA* [6].

The treatment of choice for syphilis should be benzathine penicillin G (BPG); fortunately, no TPA strains with proven resistance to penicillin have been isolated from patients, and clinical failure is rare after BPG treatment [7,8]. However, there is a major problem with the global spread of resistance to macrolides (specifically, due to azithromycin-A2058G and A2059G punctual mutations in the *23S rRNA* gene); this antibiotic group is not the first line of treatment, but is likely common due to the convenience of administration [9,10]. Additionally, the absence of moving elements associated with horizontal gene transfer mechanisms such as plasmids, bacteriophages, and transposons in the bacterium indicates that macrolide resistance is chromosomal and that its spread is due to the sexual transmission of resistant TPA strains; this possibility seemed very unlikely but is currently a global challenge [2,3,7,9].

According to the Pan American Health Organization (PAHO), in 2022, the syphilis prevalence in Latin America was estimated at 4.6 million cases [11]. The incidence of syphilis has increased in Mexico, particularly in men aged 20–24 and 25–44 years [1]. The most recent national data for MSM in Mexico reveal a syphilis seroprevalence of 15.2%, but information concerning the antimicrobial resistance of TPA in Mexicans is lacking [12].

Macrolide-resistant syphilis has been reported in Latin America, so it is assumed that Mexico will not be exempt from this phenomenon. Due to the above, this work aims to detect the resistance of *T. pallidum* to macrolides in blood samples from Mexican men who have sex with men and people living with HIV using molecular methods.

## 2. Materials and Methods

### 2.1. Samples

A bank of biological samples was generated from whole blood samples collected for projects at Clínica Especializada Condesa (CEC), CDMX (2022), and CAPASITS, Cuernavaca (2021), whose primary objective was the detection of antibodies against TPA. Both studies had a cross-sectional design. The bank collected TPA-seropositive samples from asymptomatic individuals with a VDRL titer ≥ 1:8 and a minimum volume of 200 µL; these three parameters were established as the eligibility criteria. Informed consent was obtained from the participants and confidentiality of the data was maintained. This work has the approval of the institutional ethics committee, Instituto Nacional de Salud Pública CI:1793.

### 2.2. DNA Extraction and Validation

DNA was isolated from 100 µL of whole blood via one of two methods: the first method involved a commercial ion exchange column kit (Blood Isolation MiniKit, NORGEN Biotek Corporation, Thorold, ON, Canada^®^), and the second was the phenol–chloroform–isoamyl alcohol method. The obtained DNA was measured via spectrophotometry, and the quality was evaluated via the PCR amplification of a fragment of the human beta-globin gene (140 bp) [13].

### 2.3. Molecular Detection of TPA

To identify TPA-positivity, all the samples were screened via a nested PCR protocol targeting the *Tpn47* gene [6,14]. The outer fragment was 210 bp, and the inner fragment was 135 bp. The PCR amplification conditions were modified from those of Pinilla and collaborators as described below. Briefly, the PCR mixture concentrations for a 20 µL reaction were as follows: [1X] KCl buffer, [3 mM] MgCl_2_, [0.2 mM] dNTPs, [0.32 µM] each primer, [2.5 U] recombinant Taq DNA Polymerase, 5 µL of DNA sample, and sterile water. The reaction conditions were an initial hold at 94 °C (5 min); followed by 35 cycles of 94 °C (30 s), 62 °C (30 s), and 72 °C (30 s); and a final extension of 72 °C (10 min). The mixture and conditions were the same for the outer and inner PCR steps, except that 5 µL of the outer fragment was used as the DNA template for the second round. The products were visualized via electrophoresis on a 1.8% agarose gel. The prevalence of TPA was calculated by taking the total number of samples processed as a reference.

### 2.4. Amplification of the 23S rRNA Gene

To determine the genetic resistance of TPA to macrolides, nested PCR was used to amplify a fragment of the bacterial *23S rRNA* gene with specificity for TPA. The outer fragment (628 bp) was amplified via a previously published primer pair [15,16,17]. For the inner fragment, a new primer pair (forward 23SintF 5′-GGTCCAGGCGACTGTTTATC-3′ and reverse 23SintR 5′-CTTCTCTGTCTCCCACCTATACT-3′) was designed based on a consensus genomic sequence of *T. pallidum* (strains Nichols and SS14) using the PrimerQuest Tool program, version RUO22-1233_001 [18]. The internal PCR product corresponds to a 377 bp fragment. Both primers exhibit optimal design characteristics: length (20–24 bp), GC percentage (>45%), difference between alignment temperatures (<1 °C), no secondary structures of interest (hairpins, self-dimers, and heterodimers > −5 kcal/mol), and a specific alignment with only *T. pallidum* [19].

The reaction mixture for both rounds of PCR contained [1X] HF buffer, [0.2 mM] dNTPs, [0.13 µM] of each primer, [0.02 U] Taq Phusion Hot Start II High-Fidelity DNA Polymerase (Thermo Fisher Scientific Baltics UAB, V.A. Graiciuno 8, Vilnius Lithuania^®^), and 2 µL of DNA sample (from whole blood or the PCR product) in a total volume of 20 µL per reaction. The PCR amplification conditions were an initial hold at 98 °C (30 s); followed by 30 cycles of 98 °C (10 s), 60 °C (30 s), and 72 °C (30 s); followed by a final extension of 72 °C (5 min). The products were visualized via electrophoresis on a 1.8% agarose gel and purified with the GeneJET Gel Extraction Kit (Thermo Fisher Scientific Baltics UAB, V.A. Graiciuno 8, Vilnius Lithuania^®^) for subsequent Sanger sequencing.

### 2.5. Azithromycin Genetic Resistance: Sequencing and Bioinformatic Analysis

The Nichols and SS14 *23S rRNA* TPA strains were used as references, and their GenBank accession numbers are NR_076156.1 and NR_076531.1, respectively. A file conversion of the Sanger results was performed from .ab1 to fastq with Tracy v0.7.6 [20], and the quality of the sequences was evaluated with FastQC v0.11.9 [21] using a filter of >28 Phred scale. A merged sequence for each sample was generated via the DNA Subway [22], and the Muscle algorithm was implemented with JalView v1.8.3 to perform multiple alignments [23]. The aligned resulting sequences were evaluated to corroborate the presence of the punctual mutations A2058G and A2059G. The prevalence of macrolide resistance was calculated with the samples that presented the A2058G or A2059G mutations and about the total TPA-positive samples.

## 3. Results

### 3.1. Molecular Prevalence of Treponema pallidum

A total of 90 samples were processed, all of which were classified as having active syphilis, as shown by serology (VDRL test, antibody titers ≥ 1:8). Each sample corresponds to a single person. Most of the samples were from Capasits (70/90, 77.8%), and the rest were from Clínica Especializada Condesa (20/90, 22.2%). High-quality DNA extraction was successful, as indicated by the PCR amplification of the human beta-globin gene (140 bp). TPA DNA (*Tpn47* gene) was detected in 15 samples; therefore, the molecular prevalence of syphilis was estimated to be 16.7% (15/90), CI_95%_ 10.3–25.8%. Figure 1 shows the amplified products of the *Tpn47* gene.

### 3.2. Molecular Prevalence of Mutations Associated with Azithromycin Genetic Resistance

The correct amplification of the *23S rRNA* gene was observed for all fifteen TPA-positive samples. Samples from 11 participants were sequenced after purification; four TPA-positive samples did not reach the minimal concentration required for Sanger sequencing (Figure 2).

Through a bioinformatic analysis of the eleven *23S rRNA T. pallidum* sequences, we detected one with azithromycin genetic resistance, the punctual mutation A2058G (SUB14527389 T_PALLIDUM_PALLIDUM_MEX, PP961320) (Figure 3).

The molecular prevalence of azithromycin genetic resistance was 9.1% (1/11), CI_95%_ 0.2–41.3%. Finally, Table 1 details the characteristics and experimental results of the TPA-positive samples, and Figure 4 summarizes the sample processing flowchart and the classification of the results.

## 4. Discussion

Few studies have investigated the molecular prevalence of syphilis and azithromycin resistance in Latin America. Specifically for Mexico, this study is the first to report the detection of treponemal DNA and the A2058G/A2059G mutation [2,3].

In general, knowledge about TPA is limited because of the unique characteristics of this bacteria and the difficulty in culturing it. Furthermore, molecular analysis has also been limited due to the low TPA bacterial load in patients. Our study population included persons with active syphilis (antibody titers ≥ 1:8); however, TPA DNA was detected in only 15 blood samples. Most likely, the syphilis stage (primary, secondary, and tertiary) influences the circulating bacterial load, and although the antibody titers are high, the secondary stage is the only stage with clinical evidence of TPA distribution [24]; thus, the probability of collecting a venous blood sample containing the bacteria at this stage is greater than that of the other stages [25]. Regrettably, the syphilis stage of our participants was unknown.

The participants did not have syphilitic lesions (ulcers) at the time of sample collection. Additionally, only approximately 23% of the participants self-reported having experienced symptoms suggestive of a syphilitic process, such as rash, exanthema, or condylomata. In the case of ulcers being present, PCR has been reported to be a useful diagnostic tool for syphilis, with detection sensitivity different from that of other biological samples. For example, better sensitivity was found in smear samples from primary syphilis lesions, and moderate sensitivity was reported in blood from primary and secondary stages [4,24,26]. In agreement with previous reports, we detected a low percentage of TPA-positive samples by nested PCR in the blood samples (16.7%) from asymptomatic people, which was lower than that reported by other authors [24,27] testing samples obtained from symptomatic people with primary or secondary syphilis (26% and 36%, respectively). In fact, the principal limitation of this work is the lack of information about the clinical stages of syphilis for each participant.

On the other hand, amplification of the *polA* gene presents increased sensitivity for TPA detection, so using it for detection is highly recommended [6]. Nevertheless, in the present study, it was not possible to amplify that gene, so the detection of TPA DNA was performed by amplifying the *Tpn47* gene via nested PCR. The *Tpn47* gene has no homology with other bacterial or eukaryotic proteins; thus, the reduced number of TPA-positive samples detected via PCR could be due to the low bacterial load in the blood compared with the amplified gene [6,24,28].

For the detection of molecular resistance of TPA, nested PCR was developed to amplify the *23S rRNA* gene as described in the Materials and Methods Section. The sensitivity and specificity of the detection method were improved. As a result, all the TPA-positive samples yielded the correct amplification of *23S rRNA*. However, the concentration of recovered DNA to be sequenced was low, probably due to the poor bacterial load in the biological samples. The optimal DNA amount for Sanger sequencing ranges from 100 to 120 nanograms in total according to the service provider’s requirements [29,30]. As mentioned, although the *23S rRNA* gene was amplified in all the TPA-positive samples, not all of them had sufficient concentrations for sequencing, resulting in a loss of 26.7% (4/15). This may have led to an underestimation of the prevalence of resistance.

Despite guidelines suggesting the use of intramuscular benzathine penicillin G for the treatment of syphilis, oral azithromycin (the main representative macrolide) is a very common alternative because of its method of administration and patient accessibility [30]. In approximately 2002, some treatment failures with azithromycin among MSM with syphilis were detected in San Francisco, with the A2058G mutation of the *23S rRNA* gene being the cause of resistance [31]. A report based on samples from Seattle, San Francisco, Baltimore, and Ireland (1998–2003) reported a prevalence of mutation ranging from 11 to 88% [17,31]. More recently (2009), the A2059G mutation, which confers macrolide resistance but with a lower prevalence than A2058G, was also identified [17]. Since then, A2058G/A2059G mutations have rapidly spread throughout the world [2,3,31]. The prevalence of azithromycin resistance mutations is high, nearly 90%, with wide regional variations. For example, Cuba, the USA, Canada, China, Japan, Australia, and many European countries report high prevalence rates (>61%), whereas The Czech Republic, Russia, South Africa, Madagascar, Taiwan, Peru, Argentina, and the Brazilian Marajo Archipelago report prevalences under 16% [2,3,16,32]. In a previously published meta-analysis, we reported a global increase in azithromycin resistance (A2058G/A2059G mutations) over time, and the data were stratified by mutation and continent. America showed a proportion of 52% and 4% A2058G and A2059G prevalence, respectively. Unfortunately, after a systematic review of the literature, no published data on resistance to azithromycin were found for patients with syphilis from Mexico [3]. Therefore, molecular techniques to detect TPA DNA and TPA resistance to azithromycin were developed to generate information for Mexico. The unique resistant sample identified coincides with the most prevalent mutation in America (A2058G). Even though the detection of resistance was minimal (<10%), this finding enables continued study of this event in Mexico, especially in vulnerable populations such as MSM and PLWH (core group of transmission) [2,3,33].

## 5. Conclusions

TPA DNA and resistance to azithromycin were detected in the blood samples of people with asymptomatic syphilis via molecular methodologies. This is the first report of genotypic resistance to azithromycin in *Treponema pallidum* in Mexico, which confirms the circulation of resistant TPA strains in our country, especially in vulnerable populations. The presence of resistance to azithromycin in *T. pallidum* could be a reflection of the misuse of antibiotics to treat cases of syphilis, since, at the moment, no resistance to the first-line treatment, benzathine penicillin G, has been documented.

Resistance testing for syphilis is not routinely performed in Mexico, and azithromycin continues to be prescribed as a treatment; thus, the surveillance of cases of syphilis treatment failure is recommended, especially in MSM and PLWH. These vulnerable groups contribute to the transmission of TPA and could constitute the core group for the dissemination of resistant strains. Additionally, further research on syphilis with molecular tools could improve the understanding of the dynamic evolution of antibiotic resistance and provide information on new and strengthened prevention strategies that contribute to the control of the re-emergence of syphilis, but above all, limit the dissemination of resistant TPA strains.

## Figures and Tables

**Figure 1 microorganisms-13-01069-f001:**
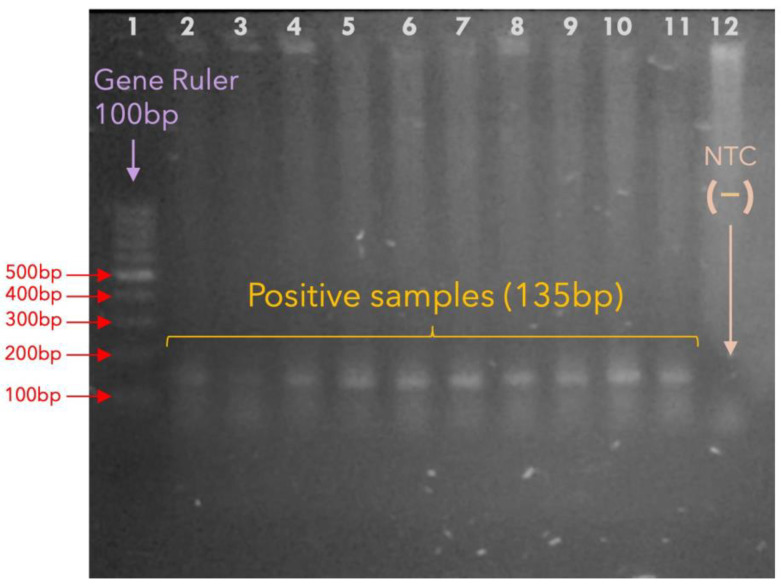
**Molecular detection of TPA.** The figure shows a set of TPA-positive samples, with the correct amplification of the *Tpn47* gene via nested PCR (135 bp). Lane 1: Gene Ruler of 100 bp; the most prominent band corresponds to 500 bp. Lanes 2–11: *Tpn47*-positive samples as indicated by the bands between the 100 and 200 bp bands of the gene ruler. Lane 12: NTC, no-template control (sterile water). The 15 positive samples were used to estimate the molecular prevalence of TPA.

**Figure 2 microorganisms-13-01069-f002:**
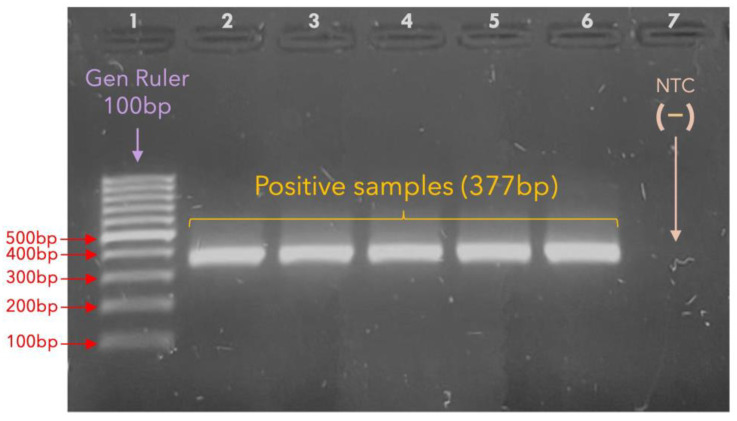
**Amplification of the *23S rRNA* gene.** The agarose gel shows TPA-positive samples with the successful amplification of the *23S rRNA* gene via nested PCR (377 bp). Lane 1: Gene Ruler of 100 bp. Lanes 2–6: *23S* positive samples, as indicated by the bands between the 300 and 400 bp bands of the gene ruler. Lane 7: NTC, no-template control (sterile water). The *23S rRNA* was amplified from all the TPA-positive samples, and they were subsequently purified.

**Figure 3 microorganisms-13-01069-f003:**
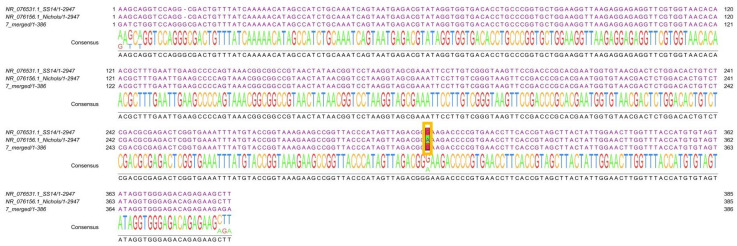
**Aligned sequence with the A2058G mutation.** Samples from 11 participants were sequenced after purification, and through bioinformatic analysis, sample 76 from CAPASITS was determined to have azithromycin genetic resistance (punctual mutation A2058G). The alignment first shows the SS14 *T. pallidum* strain (NR_076531.1_SS14/1-2947) with the A2058G mutation, followed by the Nichols strain (wild type, NR_076156.1_Nichols/1-2947), which was used as a reference, and finally, the sequence obtained from the 76 CAPASITS samples (7_merged/1-386, SUB14527389 T_PALLIDUM_PALLIDUM_MEX, PP961320). Between positions 242 and 362 (as shown at the sequence edges), the A2058G punctual mutation cab be observed (highlighted in the yellow box on the sequence). A substitution of an adenine “A” (in green) by a guanine “G” (in red) is evident. This latter base (G) indicates the presence of the mutation.

**Figure 4 microorganisms-13-01069-f004:**
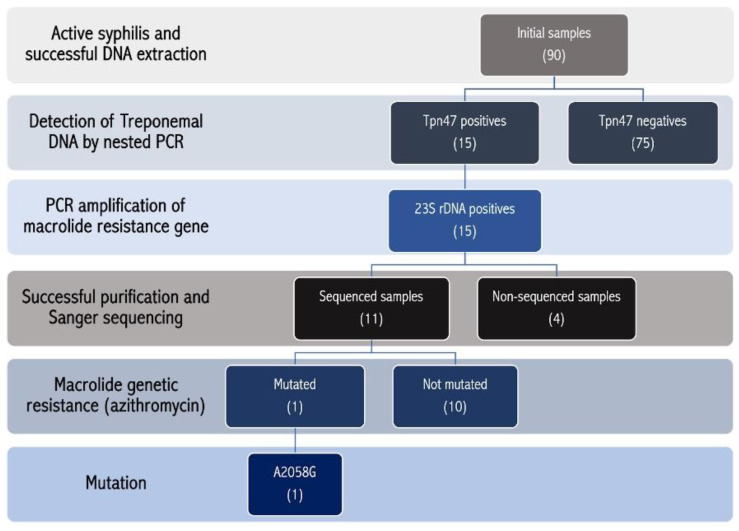
Flowchart of the study samples. Ninety samples were included in the study; for fifteen samples, the *Tpn47* gene was correctly amplified, and the *23S rRNA* gene was amplified from all the samples. Only 11 samples reached the concentration necessary to enable sequencing. One sample presented the A2058G mutation of azithromycin genotypic resistance.

**Table 1 microorganisms-13-01069-t001:** Characteristics of the TPA-positive samples. A total of 8/15 samples were from Clínica Especializada Condesa (CEC), and 7 were from CAPASITS. All the samples were classified as positive for active syphilis on the basis of their antibody titers (≥1:8). Only two TPA-positive samples were extracted via the phenol–chloroform–isoamyl alcohol method (P/C). The concentration of the DNA obtained had a very large variation [DNA 3.35-189.65 ng/µL], whereas for the purified *23S rRNA* product, only when the DNA concentrations were above [7.0 ng/µL] was there enough for sequencing (11/15). Sample 76 from CAPASITS contains the A2058G mutation (highlighted in bold), which confers azithromycin resistance. V.D.R.L., Venereal Disease Research Laboratory; ng/µL, nanograms per microliter; NM, not mutated.

No. Sample	Origin	Antibody Titers (VDRL)	DNA Extraction Method	[DNA] ng/µL	[DNA] of Purified 23S rRNA, ng/µL	Mutation
4	**C**CECC	1:32	Norgen Kit	48.30	---	NM
14	CEC	1:8	Norgen Kit	7.85	7.80	NM
15	CEC	1:32	Norgen Kit	11.15	9.90	NM
16	CEC	1:64	Norgen Kit	8.95	8.30	NM
17	CEC	1:32	Norgen Kit	3.55	7.30	NM
18	CEC	1:8	Norgen Kit	6.60	---	NM
19	CEC	1:16	Norgen Kit	8.65	39.10	NM
20	CEC	1:16	Norgen Kit	12.75	48.10	NM
25	Capasits	>1:8	Norgen Kit	4.20	47.90	NM
26	Capasits	>1:8	Norgen Kit	3.35	7.80	NM
27	Capasits	>1:8	Norgen Kit	8.15	---	NM
58	Capasits	>1:8	Norgen Kit	10.05	---	NM
61	Capasits	>1:8	Norgen Kit	78.60	51.90	NM
76	Capasits	>1:8	P/C	189.65	23.40	**A2058G**
90	Capasits	>1:8	P/C	160.80	7.00	NM

## Data Availability

7_merged/1-386, SUB14527389 T_PALLIDUM_PALLIDUM_MEX, PP961320.

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
