# Peer review of "First Report of Genetic Resistance to Azithromycin in Treponema pallidum from Blood Samples Among Men Who Have Sex with Men and People Living with HIV from Mexico"

_microorganisms, 2025, doi:10.3390/microorganisms13051069_

Round 1

Reviewer 1 Report

Comments and Suggestions for Authors

The aim of the study is plain, understandable and well formulated: to determine the occurrence of azithromycin point mutations- A2058G and A2059G- in the 23S rRNA gene of T. pallidum, from blood samples of MSM and PLwH in the homeland of the Authors.  The development and spread of azithromycin-resistant T. pallidum strains is well known worldwide, also in South America, however there are no data from Mexico.

The structure of the Manuscript is well-proportioned, the methods are adequate, conclusions are appropriate. Literature data are satisfactory. Eventually, they confirmed the presence of chromosomally resistant azithromycin T. pallidum strain in 1/11 cases.  

Though this antibiotic is not the first line of treatment, the question is of significance, as,  due to convenience, azithromycin treatment of syphilitic infection is in use, despite international data of treatment failures. This work may improve clinical practice.

They draw readers' attention to limitations of their work: low concentration of recovered DNA, and they worked with the Tpn47 gene, which does not present such sensitivity as the polA gene.

They give an acceptable explanation for these limitations.  

The aim of the study is plain, understandable and well formulated: to determine the occurrence of azithromycin point mutations- A2058G and A2059G- in the 23S rRNA gene of T.pallidum, from blood samples of MSM and PLwH in the homeland of the Authors.  

The development and spread of azithromycin-resistant T. pallidum strains is well known worldwide, also in South America, however there are no data from Mexico.

The structure of the Manuscript is well-proportioned, the methods are adequate, conclusions are appropriate. Literature data are satisfactory.

Eventually, they confirmed the presence of chromosomally resistant azithromycin T. pallidum strain in 1/11 cases.  

Though this antibiotic is not the first line of treatment, the question is of significance, as due to convenience, azithromycin treatment of syphilitic infection is in use, despite of international data of treatment failures. This work may improve clinical practice.

They draw readers' attention to limitations of their work: low concentration of recovered DNA, and they worked with  the Tpn47 gene, which does not present such sensitivity as the polA gene.

They give an acceptable explanation for these limitations.  

Author Response

COMMENTS:

The aim of the study is plain, understandable and well formulated: to determine the occurrence of azithromycin point mutations- A2058G and A2059G- in the 23S rRNA gene of T.pallidum, from blood samples of MSM and PLwH in the homeland of the Authors.  

The development and spread of azithromycin-resistant T. pallidum strains is well known worldwide, also in South America, however there are no data from Mexico.

The structure of the Manuscript is well-proportioned, the methods are adequate, conclusions are appropriate. Literature data are satisfactory.

Eventually, they confirmed the presence of chromosomally resistant azithromycin T. pallidum strain in 1/11 cases.  

Though this antibiotic is not the first line of treatment, the question is of significance, as due to convenience, azithromycin treatment of syphilitic infection is in use, despite of international data of treatment failures. This work may improve clinical practice.

They draw readers' attention to limitations of their work: low concentration of recovered DNA, and they worked with  the Tpn47 gene, which does not present such sensitivity as the polA gene.

They give an acceptable explanation for these limitations.  

RESPONSE:

We appreciate all your comments on the article we submitted. As you rightly pointed out, the study was planned in a very structured manner and, fortunately, it was successfully carried out as intended. Our interest is to continue along this line of research; however, at this stage, we are pleased to contribute molecular evidence of azithromycin resistance in circulating T. pallidum strains, which, to the best of our knowledge, represents the first such report in our country.

Reviewer 2 Report

Comments and Suggestions for Authors

Dr. Vergara-Ortega and colleagues describe an isolate of Treponema pallidum in Mexico that was found to be azithromycin resistant, adding to the current molecular epidemiology of syphilis

The manuscript is sufficiently well written, and just some relatively minor changes are needed in my opinion to make it a sound piece of work

  1. Men-Sex-Men in the title should be just MSM or spelled out properly. as it is, it simply deters interest in reading
  2. line 20. not sure what "standardization" means here.
  3. line 24. "in MSM" is not really appropriate. the point is that resistance was detected. in this case it happened to be detected from a sample coming from an MSM. As written it seems to convey a different concept
  4. line 43 "I suggest to avoid using absolute expressions like "impossible to obtain". Also the window with bacteremia is limited. this should be mentioned
  5. line 45. polA and Tp47 are mostly picked based on their specificity. the sensitivity is different based on the primers used. I suggest to rephrase saying that those are the most common targets.
  6. line 50, again azithromycin resistance is a problem that trascends MSM or PLHIV
  7. Figure 1 lacka a positive control. please provide a new figure 1 with a positive control clearly visible
  8. Figure 3 is barely visible, and likely not necessary
  9. Line 195: the statement that most known aspects of TP biology are derived from other spirochetes is not correct. please revise.
  10. line 200: II syphilis is the stage where the effect of dissemination become clinically evident, but dissemination occurs throughout the early stages. So the statement should be revised
  11. the authros say that the patients did not have I lesions. So, data on whether other kind of lesions were present (II disseminated rash, condilomata) should also be available. At least the authors should postulate whether these patients were in a latent stage.

Author Response

GENERAL COMMENT:

The manuscript is sufficiently well written, and just some relatively minor changes are needed in my opinion to make it a sound piece of work.

COMMENT 1: Men-Sex-Men in the title should be just MSM or spelled out properly. as it is, it simply deters interest in reading.

RESPONSE 1: We appreciate the observation and have replaced the term "Men-sex-Men" with the full description "Men who have Sex with Men."

COMMENT 2: line 20. not sure what "standardization" means here.

RESPONSE 2: Your concern is understandable, and we appreciate you bringing it to our attention. We have removed the term “standardization” from the abstract (line 21). The reason for using this term was to convey that we standardized the PCR methodologies we employed, meaning that we did not merely replicate them based on information from other authors, but rather adapted and validated them ourselves.

COMMENT 3: line 24. "in MSM" is not really appropriate. the point is that resistance was detected. in this case it happened to be detected from a sample coming from an MSM. As written it seems to convey a different concept.

RESPONSE 3: Once again, we appreciate you pointing out the confusion caused by the mention of the MSM population in line 24. As you rightly noted, the key point is the detection of resistance. We have removed the term to avoid any further misunderstanding.

COMMENT 4: line 43 "I suggest to avoid using absolute expressions like "impossible to obtain". Also the window with bacteremia is limited. this should be mentioned.

RESPONSE 4: Your comment and suggestion are highly valuable, the expression in question has been removed. Additionally, we have revised the idea we intended to convey and have also added information regarding the bacteremia window in infections caused by T. pallidum (lines 42-44). Reference: AEDV Expert Consensus for the Management of Syphilis. Actas Dermo-Sifiliográficas, Volume 115, Issue 9, October 2024, Pages T896-T905. L. Fuertes de Vega, J.M. de la Torre García, J.M. Suarez Farfante, M.C. Ceballos Rodríguez (https://www.actasdermo.org/es-documento-expertos-aedv-el-manejo-articulo-S0001731024003399).

COMMENT 5: line 45. polA and Tp47 are mostly picked based on their specificity. the sensitivity is different based on the primers used. I suggest to rephrase saying that those are the most common targets.

RESPONSE 5: Thank you for the correction; it is very accurate. As you suggested, we have paraphrased the sentence (line 47).

COMMENT 6: line 50, again azithromycin resistance is a problem that transcends MSM or PLHIV.

RESPONSE 6: Thank you very much for the observation; we have corrected the wording accordingly.

COMMENT 7: Figure 1 lack a positive control. please provide a new figure 1 with a positive control clearly visible.

RESPONSE 7: We appreciate your comment. Indeed, Figure 1 does not include a positive control. We initiated this project specifically by attempting to obtain a positive PCR amplification of the Tpn47 gene of T. pallidum from a group of samples with active syphilis, confirmed by VDRL testing in our laboratory. Although we did not have access to a previously confirmed positive control, the design and analysis of the oligonucleotides used were based on an alignment of T. pallidum pallidum sequences annotated in the NCBI database. Additionally, as an initial screening criterion for the samples analyzed in this project, we included only those with VDRL antibody titers ≥ 1:8, which is interpreted as indicative of active syphilis. Finally, following PCR amplification, each fragment (positive band) was purified and sequenced. The resulting sequences were then subjected to a BLAST alignment, through which the samples in question were confirmed as T. pallidum pallidum. It is for the reasons outlined above that Figure 1 does not include a positive control; however, it does present a negative control with no amplification to confirm that there was no contamination of the samples during PCR, along with evidence of processed samples showing positive amplification at the expected band size (135pb).

COMMENT 8: Figure 3 is barely visible, and likely not necessary.

RESPONSE 8: We appreciate the observation; however, with all due respect, we would prefer to retain Figure 3, as its purpose is to highlight the A2058G mutation identified in the sequence we obtained, which is of particular interest to our study. To improve the visualization of this mutation, we have enlarged the specific region of the image, thereby drawing more focused attention to the position and the corresponding nucleotide change.

COMMENT 9: Line 195: the statement that most known aspects of TP biology are derived from other spirochetes is not correct. please revise.

RESPONSE 9: We greatly appreciate your timely observation. We have removed the statement.

COMMENT 10: line 200: II syphilis is the stage where the effect of dissemination become clinically evident, but dissemination occurs throughout the early stages. So, the statement should be revised.

RESPONSE 10: We are grateful for your precision. We may not have expressed the idea clearly, but we agree that clinical evidence of dissemination appears during the secondary stage. We have revised the text accordingly (line 202).

COMMENT 11: the authors say that the patients did not have I lesions. So, data on whether other kind of lesions were present (II disseminated rash, condilomata) should also be available. At least the authors should postulate whether these patients were in a latent stage.

RESPONSE 11: Thank you for your request. In response, we have added this information in lines 207–209. We briefly note that a small percentage of participants self-reported having experienced some presumptive sign of syphilis.